

# Progress and challenges in glacial lake outburst flood research (2017-2021): a research community perspective

Adam Emmer[1], Simon K. Allen[2,3], Mark Carey[4], Holger Frey[2], Christian Huggel[2], Oliver Korup[5,6],

Martin Mergili[1], Ashim Sattar[2], Georg Veh[5]; Thomas Y. Chen[7], Simon J. Cook[8,9], Mariana Correas-Gonzalez[10], Soumik Das[11], Alejandro Diaz Moreno[12], Fabian Drenkhan[2,32], Melanie Fischer[5], Walter W. Immerzeel[13], Eñaut Izagirre[14], Ramesh Chandra Joshi[15], Ioannis Kougkoulos[16], Riamsara Kuyakanon Knapp[17,18], Dongfeng Li[19], Ulfat Majeed[20], Stephanie Matti[21], Holly Moulton[22], Faezeh Nick[13], Valentine Piroton[23], Irfan Rashid[20], Masoom Reza[15], Anderson Ribeiro de Figueiredo[24], Christian Riveros[25,26]; Finu Shrestha[27], Milan Shrestha[28], Jakob Steiner[27], Noah Walker-Crawford[29], Joanne L. Wood[30], Jacob C. Yde[31]

[1]Institute of Geography and Regional Science, University of Graz, Graz, Austria
[2]Department of Geography, University of Zurich, Zurich, Switzerland
[3]Institute of Environmental Science, University of Geneva, Geneva, Switzerland
[4]Environmental Studies Program and Geography Department, University of Oregon, Eugene, USA
[5]Institute of Environmental Science and Geography, University Potsdam, Potsdam, Germany
[6]Institute of Geosciences, University of Potsdam, Potsdam, Germany
[7]Columbia University, New York, USA
[8]Geography and Environmental Science, University of Dundee, Dundee, UK
[9]UNESCO Centre for Water Law, Policy and Science, University of Dundee, Dundee, UK
[10]Insituto Argentino de Nivología Glaciología y Ciencias Ambientales (IANIGLA) – CONICET, UNCUYO, Gobierno de Mendoza, Mendoza, Argentina
[11]Center for the Study of Regional Development, JNU, New Delhi, India
[12]Reynolds International Ltd., Mold, UK
[13]Faculty of Geosciences, University Utrecht, Utrecht, The Netherlands
[14]Department of Geology, University of the Basque Country UPV/EHU, Leioa, Spain
[15]Department of Geography, Kumaun University, Nainital, India
[16]Department of Science and Mathematics, The American College of Greece, Greece
[17]Department of Social Anthropology, University of Cambridge, Cambridge, UK
[18]Department of Culture and Oriental Languages (IKOS), University of Oslo, Oslo, Norway
[19]Department of Geography, National University of Singapore, Singapore, Singapore
[20]Department of Geoinformatics, University of Kashmir, Srinagar, India
[21]University of Iceland, Reykjavík, Iceland
[22]Environmental Studies Program, University of Oregon, Eugene, USA
[23]University of Liege, Liege, Belgium
[24]Federal University of Rio Grande do Sul, Porto Alegre, Brasil
[25]Instituto Nacional de Investigación en Glaciares y Ecosistemas de Montaña (INAIGEM), Lima, Peru
[26]National Agrarian University La Molina, Lima, Peru



[27]International Centre for Integrated Mountain Development (ICIMOD), Káthmándú, Nepal
[28]School of Sustainability, Arizona State University, Tempe, USA
[29]University College London, London, UK
[30]Centre for Geography and Environmental Science, University of Exeter, Exeter, UK
[31]Western Norway University of Applied Sciences, Bergen, Norway
[32]Geography and Environmental Studies, Pontificia Universidad Católica del Perú, Lima, Peru

*Correspondence to*: Adam Emmer (adam.emmer@uni-graz.az or aemmer@seznam.cz)

**Abstract.** Glacial lake outburst floods (GLOFs) are among the most concerning consequences of retreating glaciers in mountain ranges worldwide. GLOFs have attracted significant attention amongst scientists and practitioners in the past few decades, with particular interests in the physical drivers and mechanisms of GLOF hazard, and socioeconomic and other human-related developments that affect vulnerability to GLOF events. This increased research focus on GLOFs is reflected in
the gradually increasing number of papers published annually. This study offers an overview of recent GLOF research by analysing 570 peer-reviewed GLOF studies published between 2017 and 2021 (Web of Science and Scopus databases), reviewing the content, geographical focus as well as other characteristics of GLOF studies. This review is complemented with perspectives from the first GLOF conference (7-9 July 2021, online) where a global GLOF research community from major mountain regions gathered to discuss the current state of the art of integrated GLOF research. Therefore, representatives from
17 countries identified and elaborated trends and challenges and proposed possible ways forward to navigate future GLOF research, in four thematic areas: (i) understanding GLOFs – timing and processes; (ii) modelling GLOFs and GLOF process chains; (iii) GLOF risk management, prevention and warning; (iv) human dimensions of GLOFs and GLOF attribution to climate change.

## 1 Introduction

Sudden releases of water from a glacial lake – Glacial Lake Outburst Floods (GLOFs) – have become emblematic symptom of climate change in many mountain areas throughout the world (Clague et al., 2012; Harrison et al., 2018). GLOFs are described as 'low frequency, high magnitude events' with major geomorphic consequences (Costa and Schuster, 1988; Evans and Clague, 1994; Clague and Evans, 2000), extreme hydrological characteristics (Richardson and Reynolds, 2000; Cenderelli and Wohl, 2003; Cook et al., 2018) and possibly adverse impacts on societies (Carey, 2005; Huggel et al., 2015; Carrivick and
Tweed, 2016). While more than 1,300 historical GLOFs have been catalogued throughout the world by Carrivick and Tweed (2016), recent studies show that this number is likely a lower bound for many regions (Emmer, 2017; Nie et al., 2018; Veh et al., 2019; Zheng et al., 2021a; Emmer et al., 2022). Recent work by Veh et al. (2022) compiled a dataset of more than 2,800 GLOFs globally. These studies indicate that GLOFs may be more frequent than previously thought (Carrivick and Tweed, 2016; Emmer et al., 2022; Veh et al., 2022).





Research on GLOFs has been rapidly growing in recent decades (Emmer, 2018), driven in part by the urgent need to improve understanding trends in GLOF occurrence under climate change, and its links to retreating glaciers and the formation of hundreds to thousands of new lakes globally (Clague and O'Connor, 2015; Harrison et al., 2018; Shugar et al., 2020). These dynamics competee with increasing urbanization, land and water demand, migration, mountain tourism, and other socioeconomic and human-related forces that increase human exposure and raise vulnerability to GLOFs, especially in low

income countries such as Peru or Nepal (Carey, 2010; Sherry et al., 2018; Motschmann et al., 2020a; Sherpa et al., 2020; Carey et al., 2021). However, possible synergies and trade-offs between climate change adaptation (Moulton et al., 2021; Aggarwal et al., 2021), sustainable water use (Drenkhan et al., 2019; Haeberli and Drenkhan, 2022), hydropower generation (Schwanghart et al., 2016), glacier protection (Anacona et al., 2018) and GLOF hazard mitigation are still under discussion.

Our goal is to provide a state-of-the-art review of GLOF research for charting future research directions. We provide insights

into the GLOF research community and trends gained from a bibliometrical analysis and the first conference on GLOFs (7-9[th] July 2021, online, convened by: University of Graz, Austria; the University of Potsdam, Germany; the University of Zurich, Switzerland; and the University of Oregon, USA). This paper addresses four key questions: (i) What are the characteristics of the GLOF research community and recently published (2017-2021) GLOF papers? (ii) What are current trends in GLOF research and published GLOF papers? (iii) Where are the geographical and thematic research gaps, challenges and emerging

directions in GLOF research? (iv) Where should GLOF research move next?

## 2 Data and Methods

### 2.1 WOS and Scopus databases analysis

As a first step, we conducted a scoping review to identify most relevant GLOF studies in Clarivate Analytics' Web of Science (WOS) Core collection database (www.webofscience.com), and Elsevier's Scopus database (www.scopus.com). These

databases cover selected peer-reviewed scholarly journals, books and proceedings in the fields of natural sciences, social sciences, arts and humanities (WOS, 2022), and present themselves as the most reliable, relevant and up-to-date research databases (Scopus, 2022) that are broadly used. Both WOS and Scopus databases are among the largest available databases and are broadly used for bibliometric analysis, and studying research and publishing trends across research fields (Sandstrom and van den Besselaar, 2016; Thelwall and Sud, 2016; Da Silva and Dobranszki, 2018; Martín-Martín et al., 2018; Fire and

Guestrin, 2019). Yet, these databases have several limitations: (i) they do not capture technical reports, white papers, gray literature and local and indigenous knowledge; (ii) they are strongly oriented towards English-language literature while many journals published in other languages are not indexed; (iii) they do not necessarily capture books which are the standard form of publishing in many disciplines (for instance in humanities and social sciences); and (iv) the representation of authors from different geographic regions is uneven (Mongeon and Paul-Hus, 2016).





To ensure the consistency with the analysis of the previous period 1970-2016 (Emmer, 2018), we used an identical search string:

TOPIC: (glaci* AND outburst* AND flood* OR jökulhlaup*)

The period of interest was limited to publication date 2017-2021, returning 495 results in the WOS database and 421 in the Scopus database (search performed in March 2022). In the next step, we combined the outcomes of these databases and
removed duplicates. The resulting dataset of 570 GLOF papers is further considered in the analytical part of this study. Each GLOF paper in the database is described by several qualitative and quantitative characteristics (see Tab. 1), some of which were derived automatically from the databases, while some had to be assigned manually in the second step of the dataset building procedure.

**Tab. 1: Characteristics of GLOF papers analysed in this study.**

| Characteristics | Description | Values |
|---|---|---|
| Characteristics derived from the database: | | |
| Title | Title of the paper | 570 titles |
| Abstract | Published abstracts summarizing the work | 569 abstracts |
| Author(s) | List of authors | 2,107 unique authors |
| Year published | Year of publication in the journal | 2017 to 2021 |
| Publication title | Name of the journal, proceeding, book | 213 journal titles, proceeding titles and book titles |
| Subject area / category | Scopus subject area / Web of Science category in which the publication is indexed | 15 subject areas / 35 WOS categories |
| Document type | Classification of studies according to the structure of the content | 4 document types (article, review, proceedings paper, other) |
| WOS research domain | The WOS All databases classification of indexed papers; each paper is assigned to one or more research domains | 6 research domains (Science Technology, Physical Sciences, Life Sciences Biomedicine, Social Sciences, Technology, Arts Humanities) |



| Language | Language of the paper | 5 languages (English, Chinese, Russian, Spanish, French) |
|---|---|---|
| Open Access | Access mode by which a paper was published | GLOF paper published in any kind of open access mode (gold, green, hybrid) or not |
| Manually assigned characteristics: | | |
| No. of authors | Number of authors that co-authored a GLOF paper | Single-author or co-authored GLOF paper (1 author to 21 co-authors) |
| International cooperation | Paper is written by author(s) from one country, or by authors from several countries | National or international GLOF paper |
| Geographical focus | Regional focus of a GLOF paper assigned to one, more than one, or none of 11 GLOF hotspots (Emmer, 2018) based on the titles and abstracts | One, more than one, or none of 11 GLOF hotspots (Alaska, North-American Cordillera, Tropical Andes, Southern Andes, Iceland, Greenland, Scandinavia, European Alps, Hindu Kush-Karakoram, Himalaya, Central Asia) |

Titles and abstracts of individual papers (all papers included a title and 569 out of 570 papers included an abstract) were used for qualitative content analysis. We used a free word cloud generator (www.wordclouds.com) in order to identify frequently occurring words (and so enhanced focus) among the abstracts and titles of GLOF papers. After automatic removal of general verbs, pronouns and prepositions we manually removed other irrelevant words and clustered words with identical root. Word clouds of the 50 most frequently occurring words were visualized separately for titles and abstracts. This method has been
successfully employed in characterizing and visualizing the content of large textual data sets across other scientific fields (McGee and Craig, 2012; Atenstaedt, 2017), including the geosciences (Li and Zhou, 2017; Emmer et al., 2019). Clearly, word clouds can only illustrate priorities and the choice of wording of the group under study. Science paper word clouds (including those presented in this study) can thus differ substantially from those of local communities (e.g. Gearheard et al., 2013).

## 2.2 The GLOF conference and workshop

The GLOF conference & workshop took place from 7 to 9 July 2021 (online) and was completely open access upon pre-registration and co-organized by the University of Graz (Austria), the University of Potsdam (Germany), the University of Zurich (Switzerland) and the University of Oregon (USA). The main objective was to gather researchers dealing with GLOFs



to exchange recent knowledge and progress in GLOF research as well as to identify gaps, challenges, emerging directions and ways forward in GLOF research. This conference was organized under the patronage of the scientific standing group on Glacier
and Permafrost Hazards in Mountains (GAPHAZ; www.gaphaz.org) and has been disseminated primarily via the GAPHAZ webpage, GAPHAZ mailing list consisting of about 150 contacts and through ResearchGate (www.researchgate.net/project/GLOF-conference-workshop-7-9-July-online; about 400 reads at the time of the conference).

The conference programme consisted of four conference sessions (7 and 8 July 2021), and two discussion sessions (9 July 2021). These sessions focused on the following topics: (i) understanding GLOFs – timing and processes; (ii) modelling GLOFs
and GLOF process chains; (iii) GLOF risk management, prevention and warning; and (iv) human dimensions of GLOFs and GLOF attribution to climate change. These topics were identified based on the analysis of published GLOF papers (see Section 2.1).

The timing of individual sessions facilitated access for diverse time zones, thereby allowing organizers to engage with colleagues from across the globe and across diverse disciplines such as the geosciences, environmental sciences, engineering,
social sciences, and humanities. Individual conference sessions had between 45 to 65 attendees and consisted of five presentations each (see the conference programme: www.gaphaz.org/files/GLOF_conference_programme.pdf). A total of 37 participants from 17 countries across the globe, different scientific backgrounds and career stages, took part in two moderated sessions, during which a research community perspective on trends and challenges in GLOF research were discussed.

## 3 GLOF papers published in 2017-2021

### 3.1 General characteristics

Our dataset consists of 570 GLOF papers (see Fig. 1) of which the vast majority are classified as articles (n = 481; 84.4%), 39 (6.8%) as reviews, 28 (4.9%) as conference papers, 16 (2.8%) as book chapters and the remaining 6 papers (1.1%) are classified as others (corrections, editorials). Considering WOS research domains (unavailable for papers indexed in Scopus database), the majority of papers are assigned under the Physical Sciences domain (95.9%; Earth and Planetary sciences, Geosciences
multidisciplinary, Physical Geography and Environmental Sciences), while fewer than half (45.6%) of all published papers are assigned to the Social Sciences domain, suggesting that GLOF research is currently dominated by physical science, rather than social science, research.

The journals that published GLOF research most frequently were Geomorphology (n = 31), Remote Sensing (n = 21), Quaternary Science Reviews (n = 19), Earth-Science Reviews (n = 17), Frontiers in Earth Science (n = 17) and The Cryosphere
(n = 16). Six other journals published 10 or more GLOF papers (Natural Hazards, Science of the Total Environment, Earth Surface Processes and Landforms, Journal of Glaciology, Water, Global and Planetary Change). While some of these journals are well-established in publishing GLOF research (e.g. Geomorphology, Quaternary Science Reviews), others experienced a



recent growth in publishing GLOF papers, especially MDPI journals (Remote Sensing, Water) and Frontiers publishing house
(Frontiers in Earth Science).


**Fig. 1: Basic characteristics of published GLOF papers covered by WOS and Scopus databases. (A) Published GLOF papers in each year, a share of papers available for subscribers only and a share of Open Access papers; (B) An average number of co-authors, a share of papers written by individuals and a share of papers written by international teams.**

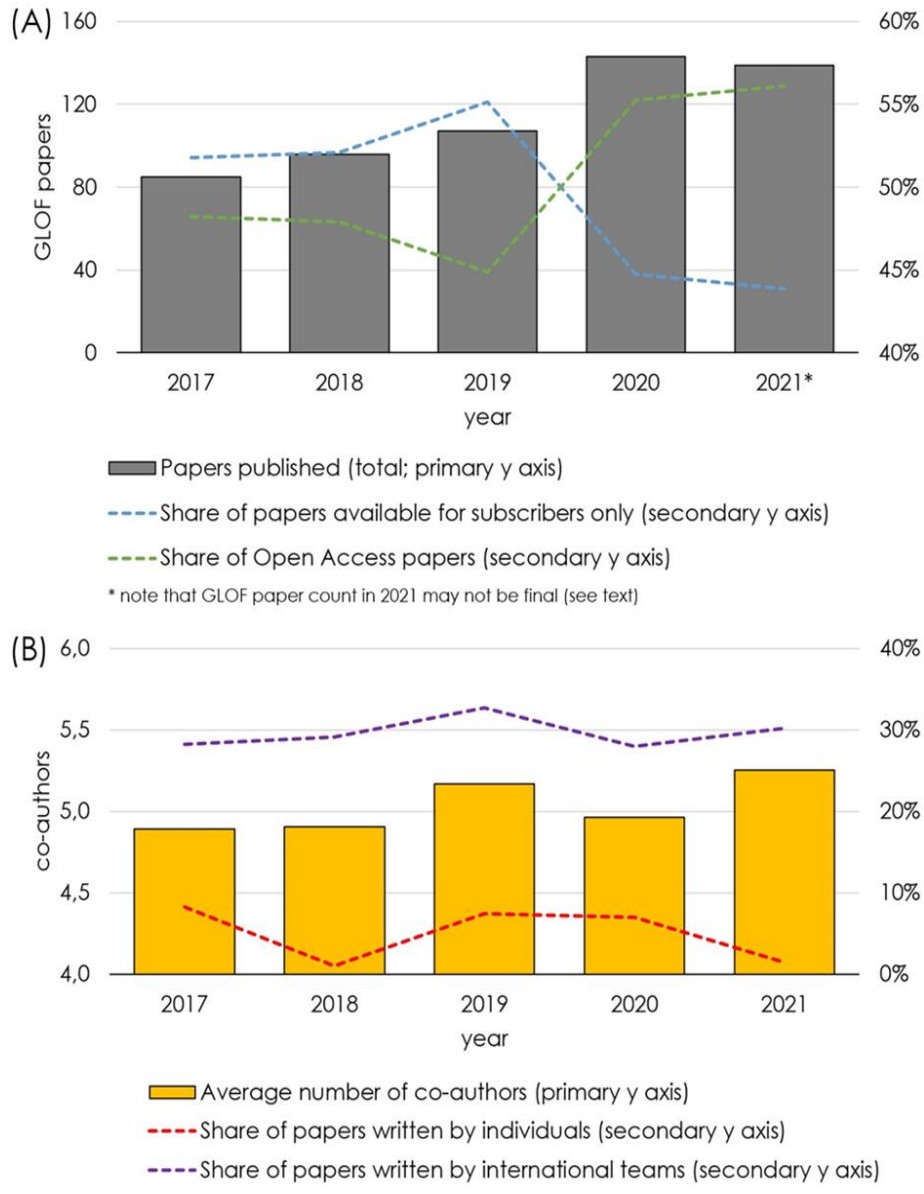



An increasing trend is observed in the number of GLOF papers published in individual years: while 85 GLOF papers were published in 2017, this number increased to 96 in 2018, 107 in 2019, 143 in 2020 and 139 in 2021 (see Fig. 1A). However, the 2021 number might not be final because of differently lagged uploads of individual journals and publishers into the WOS and Scopus databases. In comparison, Emmer (2018) identified a total of 52 papers published in 2010, 24 papers in 2000, and only 2 papers in 1990. While this is a general publishing trend (Fire and Guestrin, 2019), the number of published GLOF papers

seems to be increasing even more remarkably compared to other research fields. The gradually increasing number of GLOF papers may be explained by: (i) increasing interest of research community and funding agencies in GLOFs; (ii) growing GLOF research community; and (iii) changing publication habits (increasing need of publishing induced by changing research evaluation indicators (Emmer, 2018; Fire and Guestrin, 2019). It can also be explained by possibly expanding coverage of analysed databases. Slightly more than half of all GLOF papers (n = 292; 51.2%) were published in any kind of open access

mode, with the share varying from 44.9% in 2019 to 56.1% in 2021 (see Fig. 1A).

**3.2 Word cloud analysis - insights into study content**

The word cloud analysis is a visual representation of the most frequent words in the abstracts and titles of 570 analysed GLOF papers (Fig. 2). We grouped these most frequent words into thematic clusters: (i) glacier-related words; (ii) lake- and GLOF-related words; (iii) system- and change/dynamics related words; (iv) data-, methods- and approach-related words; (v) scale-

related words; and (vi) geographic names. Clearly, some of the recurring words may be assigned to more than one cluster (e.g., retreat* can be interpreted as a glacier-related word as well as a system- and change-dynamics related word) and are, therefore, marked in two colours in Fig. 2.

The most frequent words are related to lakes, floods and glaciers in both word clouds. Words related to data, methods and approaches (e.g., *model\**, *hazard(s)*, *assessment(s)*, *inventory* and *risk(s)*) confirmed the interest in lake and GLOF

inventorying, hazard (susceptibility, risk) assessments (Frey et al., 2018; Wang et al., 2018; Schmidt et al., 2020) and modelling of GLOFs (both back calculations and predictive; e.g., Kougkoulos et al., 2018a; Mergili et al., 2018a, 2020; Sattar et al., 2019a,b). However, studies addressing vulnerabilities to GLOFs are still rare (only mentioned in eleven abstracts and three titles; e.g., Ghosh et al., 2019), while flood vulnerability studies focusing on mountain environments in general are more common (e.g., Papathoma-Köhle et al., 2022).

The scale of GLOF studies varies from *valley-*, *river-*, *basin-*, *mountain range-* to *region*-wide, while global studies are less frequent (e.g., Harrison et al., 2018; Shugar et al., 2020). The word clouds of titles also indicate a geographical focus with dominant occurrence of *Himalaya\** (109 titles, i.e. 19.1% of all), followed by *Peru, Andes, Nepal, India, Asia* and *Greenland* (see Fig. 2). Within the system- and change-dynamics cluster, words such as *evolution* and *dynamics* are among the most frequently occurring, mostly referring to dynamics of glacier retreat and associated lake evolution (e.g., Aggarwal et al., 2017;

Kumar et al., 2020), though they allow for broader interpretations. Both word clouds contain *climate* and *change(s)*, illustrating





the overarching storyline of many GLOF studies (Harrison et al., 2018; Zheng et al., 2021c). Word cloud analysis also showed decreasing use of the word *jökulhlaup(s)* (i.e. subglacial volcanic activity-induced floods) in recent years. While up to 83.3% of published GLOF papers in 2000 – 2004 period included this keyword, this share decreased to 47.2% in the 2010 – 2015 period and to less than 25% in 2017 – 2021 period (with 11.9% share in 2021). This trend could indicate proportionally less

focus on subglacial volcanic activity-induced floods compared to other GLOF triggers and mechanisms, proportionally less focus on Iceland (Emmer, 2018), or a change in terminology towards the use of the more general term 'GLOF'.

**Fig. 2: Word clouds of abstracts (A) and titles (B) of GLOF papers. The size of individual words indicates frequency of occurrence while the background color groups individual words to thematic clusters. The 50 most frequently occurring words are shown in both word clouds.**

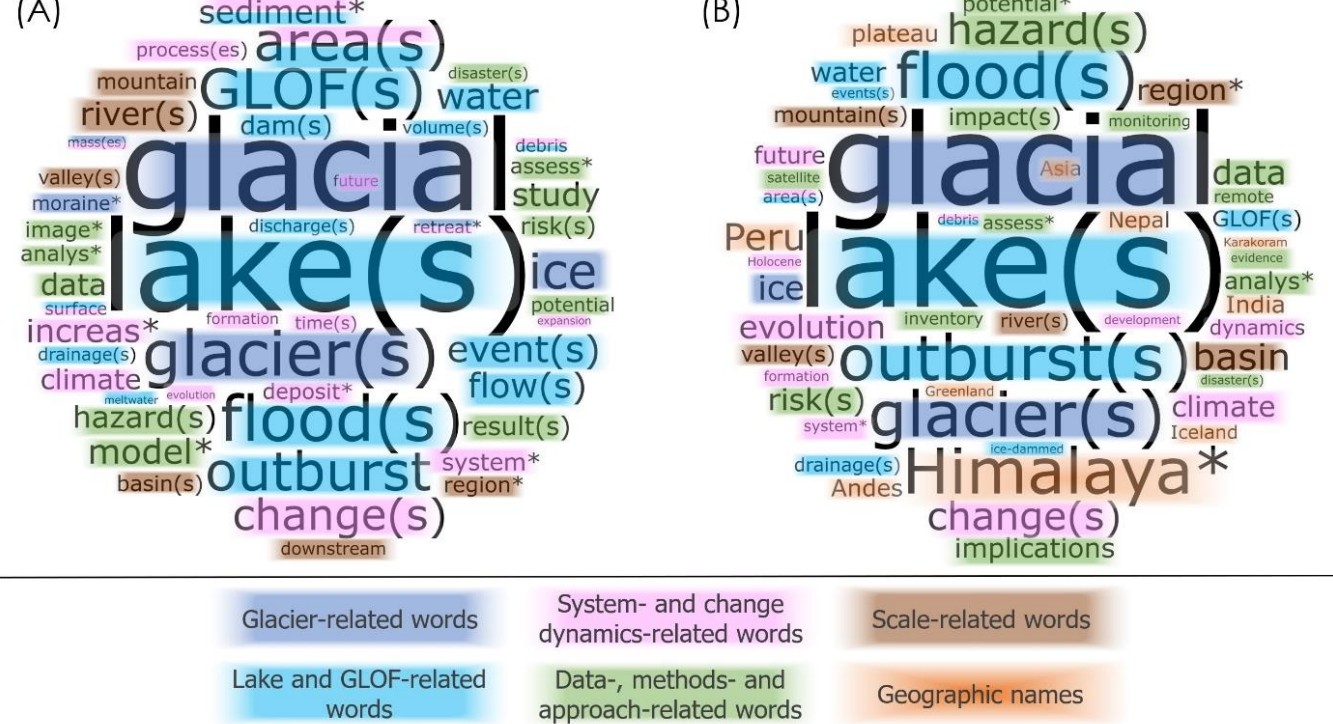


### 3.3 Geographical focus

Following previously observed trends (Emmer, 2018), a prominent hotspot of GLOF research in 2017-2021 has been in the Himalaya (HKH) – a total of 204 papers (35.8% of all) focus on this region, far exceeding the number of studies focusing on any other region (Fig. 3A). About 10% of GLOF studies focus on the European Alps, followed by Hindu Kush-Karakoram,

Tropical Andes, Iceland, Southern Andes, North-American Cordillera and Central Asia. The least studies focus on Scandinavia and Alaska. This observation is in strong contrast with the number of reported GLOFs, which show most GLOFs have occurred





in Alaska (Veh et al., 2022; Fig. 3B). Hence, the geographical focus of research appears to be driven by potential societal impacts and relevance of GLOFs and the size of these mountain regions, rather than the physical GLOF processes themselves.

**Fig. 3: Geographical focus of published GLOF papers (A), and reported GLOFs which happened from 2017 to 2021 (B) (data from Veh et al. (2022); HKK = Hindu Kush & Karakoram).**


(A)

(B)



Tens of individual GLOF events have been reported since 2017 in different parts of the world including Hindu Kush – Karakoram – Himalaya (Byers et al., 2018; Yin et al., 2019; Khan et al., 2021; Maharjan et al., 2021; Muhammad et al., 2021),

Tien-Shan (Dayirov and Narama, 2020), Tibetan Plateau (Zheng et al., 2021b), Tropical Andes (Vilca et al., 2021; Emmer et al., 2022), Southern Andes (Anyia et al., 2020; Vandekerkhove et al., 2021), the European Alps (Troilo, 2021; Ogier et al., 2021; Stefaniak et al., 2021); Alaska (Kienholz et al., 2020; Abdel-Fattah et al., 2021), mountain ranges of British Columbia (Geertsema et al., 2022), Greenland (Tomczyk et al., 2021) and Scandinavia (Andreassen et al., 2022). Recently, Veh et al. (2022) compiled globally by far the most complete GLOF inventory (total of >2,800 GLOFs in March 2022; available from:

http://glofs.geoecology.uni-potsdam.de). This compilation shows that most of the recent GLOFs have been documented in Alaska, followed by Iceland, Scandinavia (mainly Norway), the North American Cordillera and the Hindu Kush – Karakoram (Fig. 3B). Most of these GLOFs originated from ice-dammed lakes whereas GLOFs from other lake types (moraine- or bedrock-dammed) are less frequent, partly owing to the recurrence of multiple GLOFs from individual ice-dammed lakes. Repeated GLOFs from ice-dammed lakes were documented for the Karakoram (e.g. Yin et al., 2019; Khan et al., 2021), Alaska

(Kienholz et al., 2020; Abdel-Fattah et al., 2021), Southern Andes (Anyia et al., 2020; Correas-Gonzalez et al., 2020; Vandekerkhove et al., 2021), and the European Alps (Stefaniak et al., 2021).

### 3.4 Authors of GLOF papers and international cooperation

The 570 GLOF papers were published by a total of 1,960 authors, resulting in an average of about three authors per paper. However, out of these 1,960 authors, a total of 449 researchers published more than one GLOF paper each, 59 researchers

published five or more papers each and 10 researchers published ten or more papers each. The most productive 10 researchers published together about one fifth of all papers while the most productive 59 authors (3.0% of all) published all together almost two fifths of all papers. Noticeably, these papers are frequently characterized by above-average citations per year, indicating influence of a relatively small group of researchers in determining the progress direction of this growing research field.

Most GLOF papers are written by a group of co-authors, while only 4.9% are single-authored (Fig. 1B). Compared to the

previous period 1970 – 2016 analysed by Emmer (2018), this share decreased from 16.4% and is further expected to decrease considering the general declining trend in publishing single-author papers (Emmer, 2019). A GLOF paper is written by 5.05 co-authors on average. However, a tendency towards more co-authors involved in GLOF papers is observed. While on average 3.49 co-authors were involved in a GLOF paper in 1970 – 2016 (Emmer, 2018), on average 4.89 co-authors were involved in a GLOF paper published in 2017, this increased up to on average 5.25 co-authors among GLOF papers published in 2021,

possibly indicating (i) more complexity (and more interdisciplinarity) in GLOF papers being published or (ii) influence of research performance assessments (further strengthened by journal marketing), which account number of published papers but





do not take into account the number of researchers involved. Figure 1B also reveals that the share of GLOF papers written by international research teams oscillates around 30%, which is comparable to the previous period (29.1%; Emmer, 2018).

According to the WOS Core Collection Database, a total of 776 unique institutes located in 58 countries published their GLOF
research from 2017 to 2021. GLOF research was dominated by researchers affiliated with institutes located in the USA (one in four papers co-authored), followed by China (one in five papers), England, India (one in six papers each) and Switzerland (one in eight papers). Authors affiliated with institutes in these five countries all together produced about 70% of GLOF papers. Researchers from five other countries (Germany, Canada, Czechia, Norway and Pakistan) contributed to > 5% of GLOF papers each. Thirty top-productive institutes contributed to ten or more GLOF papers each. A nearly identical geographical pattern is
observed also among the papers covered by the Scopus database.

Importantly, we observe continuing trend of publications written by local researchers. While the share of Himalaya-focusing papers written by local researchers was about 15% in late 1990s, it increased to > 30% share in 2000s (+another 20% published by international teams including local researchers) and > 40% share in early 2010s (+another 15% published by international teams including local researchers; Emmer, 2018). Our study reveals that > 60% of India-focusing GLOF papers published in
2017 – 2021 were written by researchers affiliated with institutions based in India, another 20% of papers were published by international teams including Indian researchers while remaining < 20% of India-focusing papers were written by foreign researchers (without the involvement of local researchers). This is important progress because studies done by local researchers are by default often very close to government actions on disaster risk reduction and climate adaptation (see also Section 4.6).

## 4 Trends, challenges, emerging directions and ways forward

During the GLOF conference and workshop held in July 2021, we identified and discussed trends, challenges and proposed ways forward in four thematic areas of GLOF research, which we designed based on the analysis of published GLOF papers (see Section 3.2): (i) understanding GLOFs – timing and processes; (ii) modelling GLOFs and GLOF process chains; (iii) GLOF risk management, prevention and warning; and (iv) human dimensions of GLOFs and GLOF attribution to climate change. The key outcomes of these discussions are summarized in Tab. 2. Further, we elaborate in detail on the main issues
which resonated in our discussions (Sections 4.1 to 4.7). We admit that this list may be far from exhaustive; instead, we rather interpret this section as GLOF research community perspectives on the current state of GLOF research.

**Tab. 2: The summary of the key topics discussed during the first GLOF conference and workshop, observed trends, identified challenges and proposed ways forward.**

| 1) Understanding GLOFs – timing and processes |
| --- |
| Observed trends and progress: |



· Building regional to global lake inventories, including identification of future lake (see Section 4.1) and GLOF inventories (see Sections 3.3 and 4.1)

· Efforts towards better understanding of GLOF preconditions and triggers (see Section 4.2)

Identified challenges:

· Understanding lagged response of glaciers and glacial lakes to climate change signal (Harrison et al., 2018); distinguishing GLOFs associated with climate change (see also attribution part)

· Selecting and validating GLOF susceptibility indicators relevant for given geographical context (Kougkoulos et al., 2018b)

Proposed ways forward:

· Linking glacier behavior with GLOF occurrence in space and time (and triggers-oriented GLOF understanding and hazard assessment in general)

· Revision and data-driven analysis of assumed GLOF susceptibility indicators–such as rapid lake growth (Fischer et al., 2021) or earthquake GLOF triggering (Wood et al., in review)--building on recently enhanced GLOF inventories

**2) Modelling GLOFs and GLOF process chains**

Observed trends and progress:

· Substantial progress in understanding the physics of the underlying processes of GLOF initiation and propagation (see Section 4.3) such as multi-phase flows, landslide-lake interactions, entrainment and deposition, or phase separation in debris flows (Pudasaini, 2012, 2020; Pudasaini and Fischer, 2020a,b; Pudasaini and Krautblatter, 2021)

· The inclination towards integrated GLOF process chain simulations (Mergili et al., 2017; Mergili and Pudasaini, 2021) and an advancement of full 3D models capable of reproducing the relevant phenomena in a realistic way (Gaume et al., 2018; Cicoira et al., 2021)

Identified challenges:

· Obtaining (in situ) parameters required by the model (e.g., breach parameters, lake bathymetry) and addressing their uncertainties (e.g., Schaub et al., 2016; Mergili et al., 2018b, 2020; Sattar et al., 2020); see Section 4.4)

· Defining realistic GLOF scenarios in predictive GLOF modelling, accounting also for future environmental conditions (e.g., Worni et al., 2013, 2014; GAPHAZ, 2017; Sattar et al., 2021; Zheng et al., 2021c)

Proposed ways forward:





| | |
|---|---|
| · Extending a set of detail-modelled events in order to obtain a plausible range of parameters for predictive modelling (e.g., Mergili et al., 2018a,b, 2020)<br><br>· Improving availability of high-performance computers will facilitate the use of full 3D models (Gaume et al., 2018; Cicoira et al., 2021), which are currently limited by computational demands, towards operational use | |
| **3) GLOF risk management, prevention and warning** | |
| Observed trends and progress: | |
| · Unbalanced research focus on individual GLOF risk components (research of GLOF hazard still outweighs research on GLOF vulnerability and exposure, at least among the papers covered by the WOS and Scopus databases (Emmer, 2018), with relatively few integrated or holistic GLOF studies (Carey et al., 2012; Drenkhan et al., 2019, Motschmann et al., 2020b)<br><br>· Increased exposure to GLOFs across some regions in past decades due to tourism (e.g., Iceland and Peru; Matti et al., 2022a) as well as rapid and often unregulated development in GLOF-prone areas (e.g., in India and Peru; Schwanghart et al., 2016; Huggel et al., 2020a; Carey et al., 2021) | |
| Identified challenges: | |
| · Identification of overlooked or underestimated GLOF risk drivers and their consideration in GLOF risk management<br><br>· Research often does not lead to tangible actions, partly because research projects often bypass decision-making stakeholders. Scientific studies are often not well connected to the applied works of practitioners.<br><br>· Ever-growing number of lake hazard/risk assessment schemes, with different scopes (susceptibility, hazard, risk assessment) and approaches. For decision-makers, differing results and contradicting information on "potentially dangerous lakes" are difficult to interpret.<br><br>· Design and implementation of GLOF risk reduction measures and their acceptance by hazard exposed communities (see also Section 4.5) | |
| Proposed ways forward: | |
| · Interdisciplinary cooperation in GLOF research, consideration of diverse dimensions and drivers of GLOF risk beyond traditional GLOF hazard studies (see Section 4.5), connecting natural and social science communities (see Huggel et al., 2020; Carey et al., 2021; Drenkhan et al., 2019; Motschmann et al., 2020b)<br><br>· Combine disaster risk and water management under a framework of prospective lake water management with adaptive disaster risk planning considering synergies and conflicts of GLOF risk reduction measures with human water use (Haeberli and Drenkhan, 2022; Motschmann et al., 2020b) | |



| |
|---|
| · Work with local communities, decision-makers and stakeholders and their involvement in GLOF risk analysis process from the beginning (knowledge co-production, experience exchange, bottom-up and community-centered approaches; see Thompson et al. 2020; Matti and Ögmundardóttir 2021; Haeberli and Drenkhan, 2022; and Section 4.6)<br><br>- Use a consensus-based approach to identify high priority lakes for risk management, drawing across the full range of published GLOF hazard and risk studies in any given region (e.g., Mal et al. 2021) |
| **4) Human dimensions of GLOFs and attribution to anthropogenic climate change** |
| Observed trends and progress: |
| · Human dimensions are generally underrepresented in GLOF papers covered by WOS and Scopus databases, partially reflecting on publishing paradigm of individual disciplines (for instance, a tendency towards publishing books in social science domain)<br><br>· Recently increased focus on GLOF attribution to (anthropogenic) climate change (see Section 4.7), with natural scientists generally driving this research and only at times involving social science or humanities researchers (Harrison et al., 2018; Huggel et al., 2020a; Stuart-Smith et al., 2021) |
| Identified challenges: |
| · Better consideration and appreciation of cultural and spiritual values of glacier lakes and high-mountain environments, and local and indigenous knowledges in GLOF research (Lambert and Scott, 2019; Abdel-Fattah et al., 2021; Matti and Ögmundardóttir 2021; Haeberli and Drenkhan, 2022)<br><br>· More sustained, detailed work needed to understand diverse and intersecting drivers of risk and vulnerability in communities (Matti et al. 2022b; see Section 4.6), to reveal how societal variables making diverse populations vulnerable overall intertwine with GLOF risks (Carey et al., 2020)<br><br>· The politics and cultures of GLOF risk management and responses are only partially understood and analysed (Moulton et al., 2021)<br><br>· Linking GLOF occurrence patterns with anthropogenic climate change signal (Roe et al., 2021; see Section 4.7) |
| Proposed ways forward: |
| · Recognition and empowerment of (often powerful) local communities and indigenous knowledges (Mercer et al., 2010; Kelman et al., 2012; Carey et al., 2021; see Section 4.6), avoid research colonialism (come, do research and leave) and involve local researchers<br><br>· Employment of integrated, holistic, interdisciplinary approaches (Carey et al., 2012; Gall et al., 2015) and consideration of overlooked dimensions and aspects (e.g., spiritual and cultural), knowledge co-production and exchange |





· Development of robust methods (statistical methods and process-based models) for attributing GLOF occurrence and impacts to anthropogenic climate change (Stuart-Smith et al., 2021)

**4.1 Recent progress in lake and GLOF inventories**

A pronounced trend in understanding the occurrence of GLOFs from the large-scale perspective (mountain ranges, large regions) is the building of updated GLOF inventories, typically revealing incompleteness of existing GLOF records (e.g. Jacquet et al., 2017; Emmer, 2017; Carrivick and Tweed, 2019; Nie et al., 2018; Veh et al., 2019; Bat'ka et al., 2020; Zheng et al., 2021a; Emmer et al., 2022). This trend is associated with increasing availability and resolution of satellite images

(Kirschbaum et al., 2019; Taylor et al., 2021), allowing detailed analysis of persistent geomorphic GLOF diagnostic features, both in a manual as well as in a semi-automatic way (Veh et al., 2018). Geomorphic GLOF diagnostic features are frequently combined with analysis of documentary data sources (Emmer, 2017; Nie et al., 2018). More comprehensive GLOF inventories are essential for better understanding frequency of GLOF occurrence in changing mountain environments (Veh et al., 2019; Emmer et al., 2020), for revealing frequency-magnitude relationships (Hewitt, 1982; Haeberli, 1983), as well as for GLOF

attribution to anthropogenic climate change (Harrison et al., 2018; see also Section 4.7).

On a global scale, Carrivick and Tweed (2016) observed increased GLOF frequency until the 1990s, followed by a decrease in most mountain regions. The reasons for this trend, however, remained unclear. Harrison et al. (2018) observed a similar trend in outbursts from moraine-dammed lakes, possibly because of a lagged response of GLOF occurrence to climate forcing, glacier retreat, and lake formation, a concept further elaborated in the Peruvian Cordillera Blanca (Emmer et al., 2020).

However, the number of GLOFs recorded may be underestimated in some regions because of low research activity. For example, Veh et al. (2019) produced an updated GLOF inventory for the Himalayas where they observed no change in GLOF frequency since the 1980s (considering moraine-dammed lakes only). Emmer et al. (2022) prepared and updated GLOF inventory for the Tropical Andes of Peru and Bolivia, where they observed an increasing occurrence of low-magnitude GLOFs in recent decades. This trend is lake type-specific (dominance of GLOFs from bedrock-dammed lakes, reflecting later stages

of deglaciation in the Tropical Andes) and may also be biased by decreasing amount and availability of remote sensed and documentary data, and vanishing geomorphological GLOF imprints of events further back in time (Emmer et al., 2022). Most recently, Veh et al. (2022) compiled a global inventory of >2,800 GLOFs and observed a flatter trend in GLOF occurrence since the 1970s.

An emerging trend in GLOF research is tied with the identification of locations suitable for future lake formation. Considering

sustained glacier retreat under different Representative Concentration Pathways or complete deglaciation, several recent studies have attempted to locate potential future lakes and quantify their volumes, for instance in the Swiss Alps (Gharehchahi



et al. 2020), in the Austrian Alps (Otto et al., 2021), in High Mountain Asia (Furian et al., 2021; Zheng et al., 2021c), and on the global scale (Frey et al., 2019).

## 4.2 GLOF triggers and GLOF susceptibility indicators

A remaining issue is the appropriate selection of GLOF susceptibility indicators in regional GLOF susceptibility and hazard assessment studies. Kougkoulos et al. (2018b) identified 79 different GLOF indicators in previous studies. The selection of GLOF susceptibility indicators frequently relies on an expert-based analytical hierarchy process (AHP) with subjectively defined or adopted thresholds (e.g., Aggarwal et al., 2017; Muneeb et al., 2021), while statistic-based studies building on rigorous analysis of previous GLOFs are rare (McKillop and Clague, 2007a,b; Fischer et al., 2021). For a given lake,

characterising the current conditions and those prior to outburst remains challenging, highlighting the need for comprehensive lake and GLOF inventories (e.g. Petrov et al., 2017; Buckel et al., 2018; Wilson et al., 2018; Bat'ka et al., 2020; Shugar et al., 2020; How et al., 2021; Lindgren et al., 2021; Mölg et al., 2021; Wood et al., 2021; Andreassen et al., 2022; Emmer et al., 2022; Veh et al., 2022). Recent research efforts revealed that some of the broadly accepted indicators of GLOF susceptibility assessments may have ambiguous roles. An example is the control of earthquakes in triggering GLOFs. While numerous

GLOF susceptibility assessment studies consider earthquakes as possible triggers of GLOFs, recent studies showed that very few GLOFs have been actually triggered by earthquakes globally (Kargel et al., 2016; Wood et al., in prep). Another example is rapid lake growth, which is also frequently used as GLOF susceptibility indicator (see the overview of Kougkoulos et al., 2018b); however, Fischer et al. (2021) showed that this characteristic may not be an indicator of GLOF occurrence in the Himalaya.

On local scale, a recent trend goes toward better understanding of controls, preconditions, triggers of, and interactions during individual GLOFs (Carrivick et al., 2017; Blauvelt et al., 2020; Vilca et al., 2021), considering also the role of climate and climate change (Zheng et al., 2021b). Numerous studies describe, analyse and model not only the hydrodynamics and geomorphological imprints of GLOFs (e.g. Clague and Evans, 2000; Emmer, 2017; Jacquet et al., 2017), but also assess pre-GLOF conditions, hazard drivers (climatological, glaciological, geological), and elucidate plausible GLOF scenarios (e.g.,

Carrivick et al., 2017; Mergili et al., 2020; Klimeš et al., 2021; Zheng et al., 2021b; Emmer et al., 2020, 2022).

Importantly, a proper terminology should be maintained among researchers and also among disaster risk reduction practitioners and authorities in order to avoid misinterpretation of individual events. Many mass flow events are often immediately termed as GLOFs, because GLOFs have gotten so much attention recently. An example is an early interpretation of the 2021 Chamoli disaster, which was shortly after it happened described as GLOF by some. However, detailed analysis revealed that it originated

as rock and ice avalanche and no lake was involved in the process chain propagation (Shugar et al., 2021).





### 4.3 Two decades of GLOF modelling

Simulations of GLOF process chains have been performed since the early 2000s and at least three stages of research evolution can be distinguished:

1. Relatively simple empirical mass point models such as MSF (Huggel et al., 2003, 2004), mainly suited for regional-scale applications, and still applied more recently at such scales (r.randomwalk and its predecessors: Gruber and Mergili, 2013; Mergili et al. 2015).

2. More advanced model chains, applying tailored physically-based simulation tools for each component of the process chain and coupling them at the process boundaries (Schneider et al., 2014; Worni et al., 2014; Schaub et al., 2016).

3. With the emergence of two-phase (Pudasaini, 2012) and later three-phase (Pudasaini and Mergili, 2019) mass flow models and related simulation tools (Mergili et al., 2017; Mergili and Pudasaini, 2021), the trend has moved towards integrated simulations, considering the entire GLOF process chain in one single simulation step.

We now mainly focus on the latest stage, the integrated modelling of GLOF process chains, and identify three main lines of challenges: (i) defining GLOF scenarios; (ii) exploring the field of tension between the physical detail and practical applicability of the available simulation tools; and (iii) moving from successful back-calculations to reliable predictions. With regard to GLOF scenarios, it is often the worst-case scenarios which are most relevant for informing risk management. However, the decision on what are realistic worst-case scenarios for specified time horizons, and what are unrealistically apocalyptic assumptions, is sometimes disputed.

With regard to modelling the dynamics of GLOFs, we note recent progress in the underlying physical processes during lake outbursts, such as multi-phase flows, landslide-lake interactions, entrainment and deposition of sediments, or phase separation in debris flows (Pudasaini, 2020; Pudasaini and Fischer, 2020a,b; Pudasaini and Krautblatter, 2021). Also, the prevailing depth-averaged models are challenged by machine learning techniques and full 3D models, which are able to reproduce the studied phenomena in a highly realistic way (Gaume et al., 2018), but are still computationally too demanding for operational application on complex, long-runout process chains – a situation which might improve in the coming years and decades.

### 4.4 Required but hard-to-obtain modelling parameters

A particular challenge is the dependence of advanced physically based models on unknown parameters required for modelling erosion and deposition (Pudasaini and Fischer, 2020a; Pudasaini and Krautblatter, 2021). In the physically based models, it is mainly the difference between the mechanical properties of the flow and those of the basal surface which determines whether erosion or deposition of solid material occurs, resulting in an extremely high sensitivity of the model results on barely known material properties. Even though such models have been implemented at least in semi-operational software tools such as





r.avaflow (Mergili and Pudasaini, 2021), practitioners still prefer models which are more straightforward to parameterize, and where guiding parameter values for different processes and process magnitudes are available.

Such guiding parameter values would be extremely valuable for predictive simulations of future GLOF scenarios, but in contrast to "ordinary" debris flows or snow avalanches, which occur at much higher frequency, they are not yet available for GLOF process chains. While several such cascades have been successfully back-calculated in the last years (Mergili et al.,
2018, 2020; Vilca et al., 2021; Zheng et al., 2021), predicting future GLOF process chains remains a major challenge – not only in terms of defining scenarios of lake growth or volumes of possibly impacting landslides, but also in defining appropriate sets of model parameters.

With the growing need for hazard assessments in areas with limited access, where potential GLOF exposure in the downstream regions is high (Allen et al., 2016; Schwanghart et al.,2016; Allen et al., 2019), predictive GLOF modelling serves a purpose
(e.g., Sattar et al., 2019; 2021). However, such modelling demands prior evaluation of the breach parameters such as breach depth, width, and the breach formation time. These parameters are difficult to estimate and depend on multiple factors such as the nature of the damming material, the trigger event (e.g., avalanche, landslide, internal moraine failures), nature of the impact wave, freeboard of the lake and lake bathymetry. Therefore, one must rely on empirical methods or scenarios definition as alternatives to determine these parameters exactly. Although numerous empirical approaches have been developed to calculate
the breaching parameters (MacDonald and Langridge-Monopolis, 1984; Costa, 1985; Bureau of Reclamation, 1988; Von Thun and Gillette, 1990; Froehlich, 1995), a consensus on their suitability for glacial lakes has not yet been established. With the advancement of numerical modelling approaches where GLOF process chains can be efficiently modelled (e.g., r.avaflow) these parametric uncertainties of moraine breach can be minimized as breaching would depend on the kinetic energy over the entrainable material (moraine in this case) of the modelled GLOF process chain.

**4.5 GLOFs and human dimension contexts**

Given the community diversity and range of disciplines among GLOF researchers as well as the broad range of local and Indigenous knowledges, there is a wide variety of methods, questions, motivations, objectives, and framings around the GLOF problem itself. Indigenous communities may have their own knowledge about glacial lakes, and put their knowledge into larger histories of colonialism, dispossession, and racism, not to mention into larger reciprocal interactions between a fluid human
and non-human world. Natural scientists work to understand physical drivers of GLOFs and predict their impacts. Social scientists often hope their research can contribute to local empowerment, community sovereignty, self-determination, and environmental justice, which significantly transcend analyses of the water flowing downstream from, for example, an overtopped moraine dam. It is important to recognize and discuss these different underlying goals that influence not just research questions but also methodological approaches to the research and larger politics of knowledge systems. GLOF
research now increasingly recognizes the need to work with communities, co-produce knowledge, and co-manage landscapes.



At the same time, there is perhaps not enough attention to the ways in which different researchers and stakeholders fundamentally define the GLOF problem differently from the outset.

Given the existing research gaps around the human dimensions of GLOFs and the pressing need for future studies to address attribution of GLOFs, there are three key emerging trends in GLOF research that could be productively engaged and expanded:
(i) GLOF contexts, particularly for people living near glaciers; (ii) GLOF governance and the broadening of stakeholders involved in GLOF prevention, risk reduction, and management; and (iii) better understandings of GLOF attribution, to pinpoint whether anthropogenic climate change affects GLOF risk and to explore how different social groups understand cause-effect related to GLOFs. First, local communities living near ice face multiple risks including, but also beyond, GLOFs, which implies that GLOF risk needs to be seen in a more comprehensive social-environmental context. Long-standing research has
shown that communities exposed to GLOFs are diverse with respect to race, class, gender, age, religion, education, language, geographical location, and other socioeconomic variables specific to places and historical contexts (Gagné, 2019; Sherry et al., 2018; Haverkamp, 2021; Carey, 2010). Mountain communities are often distant from cities and centers of power, making them marginalized politically and neglected when it comes to infrastructure, hazard mitigation, government assistance, health care, education, and economic investments. These factors increase the vulnerability of mountain communities to GLOFs.

In many cases, mountain people have been subjected historically to intrusions by outsiders, from missionaries and mining companies to tourists and national park administrators who can restrict local access to high-mountain spaces and resources. Research thus shows that risk is distributed unequally across populations and that GLOFs are far from the only hazard communities face (Matti and Ögmundardóttir 2021; Matti et al., 2022b). GLOF studies focusing on the human dimensions must therefore recognize these other risks beyond the glaciers, from other geohazards such as earthquakes, floods and
landslides, through to food insecurity and misogyny to land loss, droughts, cold waves, and water contamination.

Yet it is also important to recognize that local communities downstream from glacial lakes are not simply victims; they do more than just struggle against perpetual and widespread risks. Many have chosen historically to live outside hazard zones, such as Peru's Indigenous communities around the Cordillera Blanca (Figueiredo et al., 2019). Others have migrated away from regions exposed to GLOFs either in response to disasters in the Andes (Wrathall et al., 2014) or interpret risk through
community-specific conceptualizations and culture, as Sherry et al. (2018) examine below Nepal's glacial lake Tsho Rolpa. Others understand that glacial lakes can generate floods, but they also deem lakes important for water storage, hydropower, and tourism (Moulton et al. 2021; Matti et al. 2022b). Still, others maintain cultural and spiritual relations with glaciers, though these have sometimes had to transform due to ice loss (Allison, 2015; Gagné, 2019). What is more, cosmology and religion are important to understand with GLOFs because religious leaders are community leaders, and because state authority and
politics flow into communities through spiritual organizations and structures (Hovden and Havnevik, 2021). Given the diversity of communities and experiences, it is crucial for GLOF researchers to analyse local populations in ways that





acknowledge this diversity without simply lumping all community members together, by recognizing the various contexts and forces that influence people near glacial lakes.

## 4.6 Participation, Management, and Governance of GLOFs

Researchers increasingly recognize that local communities should be involved in GLOF studies, climate adaptation, and disaster risk reduction policies and initiatives that could affect them and their region. Examples show that even when communities have been involved in, and supportive of, projects, conflicts can still emerge around GLOF prevention or other glacier research. In Peru, for example, local residents have resisted hazard zoning policies to prevent construction inside potential outburst flood paths. In another case, some local residents destroyed a GLOF early warning system at Laguna 513
(Huggel et al., 2020b), while others fought against glacier ice core research on Mount Huascarán. In the Everest region, there was a disconnect between the local communities and the outside agencies regarding their priorities on cryospheric hazards and risks (Sherpa et al., 2019; Thompson et al 2020). These examples highlight the importance of building relationships with local communities and engaging them with research and adaptation planning from an early stage.

Building trust with communities is key and implies a number of responsibilities, also for GLOF researchers, such as following
community-driven processes and working toward co-production of knowledge and co-management of the projects. As Carey et al. (2020) note in their concept of "glacier justice", research should be driven by communities, include multiple forms of knowledge including local and Indigenous knowledges, and recognize diverse aspects of vulnerability for communities near glaciers. Haverkamp (2021) has examined glacier-related research and adaptation work in the Peruvian Andes to show that top-down, techno-scientific, and developmentalist approaches tend to drive outsiders' projects, thereby perpetuating a form of
colonialism, intervention, or extraction. She thus calls for "adaptation otherwise" as an approach that works with and in support of highland communities so that glacier studies do not further marginalize and disempower them. The preparation of ethical guidelines for GLOF researchers working in places with communities might help to promote these efforts. Further insights are also available from other fields and environments; for instance, Holm et al. (2011) offer guidance for ethical research practices in Greenland, and explain that research should follow established institutional guidelines in the research country and the
researcher's home country. It also includes partnerships with residents of the country and community, the sharing of results in the country and communities of research, and the need to help build scientific literacy and research expertise within the country and communities of research. Furthermore, Whyte (2020) calls for research that is guided fundamentally by consent, trust, accountability, and reciprocity with local communities, particularly indigenous people.

## 4.7 Attribution of GLOF events to anthropogenic climate change

A direction of interdisciplinary research that has gained considerable traction in recent years examines the multiple drivers of risks related to GLOF events. A particularly debated question is whether anthropogenic climate change causes GLOF hazards, and if so, to what extent. Recent research has established clear causality from greenhouse gas emissions to glacier shrinkage,



lake growth and formation and, possibly to GLOF hazard (Harrison et al., 2018; Huggel et al., 2020a; Stuart-Smith et al., 2021). This research now starts to inform climate litigation. The most prominent case internationally is currently being debated

at a German court and is based on a claim of a citizen of Huaraz, Peru, who understand that the emissions of the German energy company RWE have contributed to placing his home at risk of flooding from glacier lake Palcacocha. The case has not yet been decided but a court has admitted such a case to the evidentiary stage, implying that the defendant (RWE) will be held liable for the damage or risks at a place thousands kilometers away if causality between their emissions and the (potential) damage can be established. The fact that research can inform such climate litigation cases should not prevent from putting the

issue at stake into a broader and more comprehensive perspective of responsibilities and justice. For the case of Lake Palacacocha, Huggel et al. (2020a) have analysed both climatic and non-climatic drivers of risk, including governance, social and economic conditions and development, or cultural traits that all strongly influence how people and values are exposed and vulnerable to GLOF hazards, and what type of local, national and global responsibilities are implied as a consequence.

## 5 Concluding remarks and recommendations

Our analysis of 570 GLOF papers published in 2017-2021 revealed that: (i) the amount of published GLOF papers experienced a sharp rise (+70% more papers in 5 years) and the majority of these papers were published in journals indexed primarily under geoscientific categories; (ii) a relatively small group of researchers produced a substantially large amount of influential GLOF papers (3% of the most productive researchers contributed to 40% of GLOF papers); a similarly unbalanced pattern is observed among countries and institutes involved in GLOF research; (iii) an average number of co-authors of a GLOF paper had

gradually increased, possibly indicating more interdisciplinarity and complexity in GLOF research; (iv) detailed insights from Himalaya reveal gradually increasing share of publications written by local researchers, suggesting improved chances for the acceptance of research result and the implementation of appropriate disaster risk reduction measures; (v) a prominent hotspot of GLOF research is the Himalaya region, while the majority of recent GLOFs are documented from repeated outbursts of ice-dammed lakes in Alaska, Karakoram, Iceland and Scandinavia, revealing a geographical discrepancy and potential societal

impacts as driver of GLOF research; and (v) a word cloud analysis tracked a trend towards linking GLOFs to changing climate and an upswing of modelling approaches; it also confirmed a lack of studies addressing vulnerability and exposure to GLOFs.

Discussions and insights from the first global GLOF conference and workshop, with attendance of GLOF research community members from all over the world and from various scientific background and career stages, allowed us to identify challenges and outline general recommendations for ways forward in GLOF research (see Tab. 2). To navigate future GLOF research

towards addressing identified challenges, we especially recommend: (i) to promote GLOF triggers-focusing analysis and hazard assessments, data-driven re-analysis of GLOF susceptibility indicators; (ii) to back-calculate relevant events in order to refine model parameters and define plausible sets of parameters for predictive modelling of potential future events; (iii) to foster interdisciplinary cooperation, and the employment of integrated holistic approaches in GLOF research enabling the



identification and consideration of diverse drivers, aspects and components of complex GLOF risk; and (iv) to support the
involvement of local researchers, communities, decision-makers, authorities and other stakeholders promoting knowledge co-
production and experience exchange that are fundamental for the consideration, acceptance and utilization of GLOF research
outcomes and improved future GLOF risk management.

**Author's contribution:** The idea of this work arose from a discussion among the core group of the GLOF conference
organizers (University of Graz, University of Oregon, University of Potsdam and University of Zurich). The conference was
organized under the patronage of the GAPHAZ standing group (Glacier and Permafrost Hazards in Mountains). All co-authors
contributed to the discussion and the writing process and approved the final version of this text.

**Competing interests:** The authors declare no competing interests.

**Code/Data availability:** The literature search for the scoping review was done using Scopus database (www.scopus.com) and
Web of science (WOS) database (www.webofscience.com). Data about recent GLOFs (Fig. 3) are available from:
https://glofs.geoecology.uni-potsdam.de (Veh et al., 2022). All other data generated or analysed during this study are included
in this article or available from the corresponding author on request (adam.emmer@uni-graz.at),

**Disclaimer:** The views and interpretations in this publication are those of the authors and are not necessarily attributable to
ICIMOD.

**Acknowledgement:** The authors would like to express their thanks to the scientific standing group on Glacier and Permafrost
Hazards in Mountains (GAPHAZ; www.gaphaz.org) of the International Association of Cryospheric Sciences (IACS) and
International Permafrost Association (IPA) for the conference patronage and financial support of this publication.

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
