# Peer review of "Progress and challenges in glacial lake outburst flood research (2017-2021): a research community perspective"

_Natural Hazards and Earth System Sciences, 2022_

## Author Comment (AC1)

In this study, the authors reveal the current progress and challenges of glacial lake outburst flood research in recent five years (2017-2021) launched from the first GLOF conference (7-9 July 2021, online). This analysis was based on the collections of 570 peer-reviewed GLOF studies published in 2017-2021 (Web of Science and Scopus databases). Four thematic areas related with GLOFs were summarized. This study is interesting and this manuscript was presented well. I recommend this manuscript was to publish in NHESS with the further improvement of the suggested comments below.

We would like to thank Guoqing Zhang for his review to our manuscript and overall positive evaluation of it. Below we provide our point-by-point replies (in blue).

Major comments:

1) The papers published in 2021 were collected fully? The number of papers in 2021 is smaller than 2020, the search was performed in March 2022. Some papers published in 2021 have a delay to update, especially after March 2022. I suggest the authors to update the number of papers published in 2021. In addition, some papers in Discussion/Preprint status were included in the statistics? The status of these papers can be changed and this could affect the finial results.

Yes, we built the database in March 2022, which means that not all papers published in 2021 were already loaded in Scopus and WOS databases. In the revised version of the manuscript, we will update the 2022 numbers and subsequent interpretations accordingly.

2) How about the current progress by different countries? The reader could be interesting the trend of leader authors from the different countries, especially in high mountain regions from the developing countries status?

The geography of GLOF research (both geography of institutes / authors of GLOF research as well as geographical focus of GLOF research papers) are addressed in section 3.4. In the revised version of the manuscript, more attention will be paid to stats of GLOF research papers published by the authors from developing countries, as suggested.

3) The description of statistics of published papers is clear. However, the key finding of this study is the current progress and challenges of GLOF. At present, these are mainly described in text. It is possible to add some schematic diagrams to display these directly?

Current progress and challenges in GLOF research are summarized in Table 2 and further elaborated in individual sub-sections of Section 4. A schematic diagram will be considered for the revised version of the manuscript.

Specific comments:

1) Table 1. The classifications of four thematic areas are considered in Table 1?

In fact, we do not strictly assign each paper to one or more thematic areas (these were rather used in the GLOF conference), therefore it is not mentioned in Table 1. It will be clarified in the revised version of the manuscript.

2) Figure 1 need to be improved for publication.

The quality of the figure will be checked.

3) Table 2 need to be redesigned for readable. This table is too long, and sources can be separated in a new column.

We will consider splitting this table into 4 separate tables, according to 4 thematic areas addressed in the paper.

4) Page 17, L300: "(e.g., Aggarwal et al., 2017; Muneeb et al., 2021)" suggested references here: doi: 10.3389/feart.2021.775195

Suggested references will be cited there.

5) Page 17, L305: please decrease the number of papers cited at one place. You can separate it at several places. Others need similar corrections.

Will be edited accordingly and checked throughout the manuscript.

6) Page 17, L315: "On local scale, a recent trend goes toward better understanding of controls, preconditions, triggers of, and interactions during individual GLOFs (Carrivick et al., 2017; Blauvelt et al., 2020; Vilca et al., 2021)," suggested references here: doi: 10.5194/tc-15-4145-2021; doi: 10.1017/jog.2019.13

Suggested references will be cited here.

7) Page 19, L360: Allen et al., 2016; Schwanghart et al.,2016; Allen et al., 2019 -> Allen et al., 2016, 2019; Schwanghart et al.,2016. Others need similar corrections.

Will be edited accordingly.

**Citation**: https://doi.org/10.5194/nhess-2022-143-RC1

Thank you again for reviewing our manuscript!

On behalf of the collective of authors

Adam Emmer

---

## Author Comment (AC2)

1) general comments

The consortium of authors provides an up-to-date insight into the development of scientific reporting and research related to the phenomenon of floods from glacial lake outbursts worldwide. In doing so, the comprehensive study analyzes the time period from 2017 - 2021 in a continuation of previous work, noting important trends. The study was launched on the occasion of the first GLOF conference in 2021 and is based on the evaluation of more than 500 scientific articles recorded in the authoritative scientific publication databases Web of Science and Scopus. The evaluation is very comprehensive and well structured.

We would like to thank the reviewer for their review to our manuscript. Below we provide our point-by-point replies (in blue).

However, it lacks the classic division of chapters into results, evaluation, and discussion after the introduction and data and methods. A difficulty is the consistent delimitation of the time period and the consideration of articles that have been submitted but not yet final published. Here, an identical procedure is also of great importance for future similar work.

We agree that our work does not follow standard structure of the scientific work. This is in part because the paper has authors from more than a dozen disciplines across the natural sciences, social sciences, and humanities. There is not one structure and format for papers that are interdisciplinary. Nevertheless, we'll modify that in the revised version of the manuscript in a way that it is clearly recognizable what the results are.

A glacial lake outburst flood becomes more and more of a compellingly dangerous process as the article progresses. Not sure if this impression is intentional or could be softened a bit.

This was not our intention and we'll check the wording.

A graph showing the temporal development of the published articles per region would enrich the article in chapter 4.1. The presentation of the tables could be improved and made more attractive.

We'll consider adding a figure or a paragraph showing temporal development of published articles per region. In line with the comment of Reviewer #1, table 2 will be split in 4 tables in order to increase the readability.

In the final part of the relatively long discussion, strong emphasis is placed on sensitivities of an indigenous population and hurdles of the assessors with respect to hazard communication. These aspects, while very exciting, do not necessarily belong to the paper and would be worthy of a separate publication at best.

We actually aimed at covering a broad variety of GLOF hazard and risk components and we find keeping those parts in line with objectives of our manuscript.

The paper is linguistically well written with a good reading flow. The list of references is not surprisingly very long. Here, it would be worth considering to position the entirety of the references in an appendix and to mention only the really relevant papers in the article.

Yes, the list of references is long, but it doesn't mandate us to move cited references to an appendix. We feel that papers cited in the manuscript are all relevant references and so should appear in the list of references. In fact, our goal is to offer a multidisciplinary perspective and thus retaining the diverse and extensive references is crucial toward this end.

Basically: one wishes for conferences with a similar output and such a good overview also in other related research areas!

Thank you for your overall positive evaluation of our work.

2) specific comments

Chapter 2.1 mentions the difficulty in dealing with scientific articles in languages other than English, which are not (completely) recorded in the databases. It also mentions the abundance of technical reports, gray literature, and local and indigenous knowledge. It is not clear how this will be dealt with.

Our analysis mainly builds on records of Scopus and WOS databases. However, we feel it's important to mention apparent drawbacks of those databases, as we did. Considering the scope of our study, we can't dig into technical reports, grey literature as well as local and indigenous knowledge in their full breadth.

Chapters 3 and 4 represent results. They could be named accordingly (with creation of additional subchapters).

We will rename those sections accordingly.

In Table 2, the 3rd part states the challenges in management, preparedness and warning. What is the management of non-events or good-natured glacial lake outbursts?

Unfortunately, we do not understand the point made by the reviewer in this comment.

In the same Table 2, the comment in the first section on observed Trends and Progress already seems to me to be a strong interpretation and not a neutrally presented result.

This manuscript – to a large extent – presents the views and perspectives of members of the GLOF research community. Therefore, some of the statements may not be considered neutral. However, we don't find much interpretation in 'Observed trends and progress' part of Table 2.

Maybe add a graph with number of events per region over time in chapter 4.1.

Here we refer to recent GLOF inventory compiled by Veh et al. 2022 (http://glofs.geoecology.uni-potsdam.de/).

From chapter 4.2 on, I have the impression that the text changes to a discussion of the results. At the end of the chapter, clarification of when a glacial lake outburst becomes a GLOF would be even more precise.

Unfortunately, we do not understand the point made by the reviewer in this comment.

The discussion of the human dimension context in chapter 4.5 and chapter 4.7 is relatively long at the end of the paper, but in my opinion it is not in the center of the study. Possibly shortening or even transferring it to a separate article should be considered.

Actually, we argue that tackling and integration of diverse dimensions and components of GLOFs make our manuscript unique and we thus do not plan to shorten/transfer those parts, which we consider integral for understanding GLOFs in a holistic way. A fundamental argument of the paper is that human dimensions cannot be sidelined or relegated, and we thus believe the section is essential for our article.

3) technical corrections

Figure1 should be be improved for publication

Figure 1 will be revised accordingly.

Table2 needs improvement for better readability. List references separately.

To increase the readability, table 2 will be split into 4 smaller tables.

page8, line172: number «abstracts» with (A) and number «titles» with (B). This creates the reference to Fig. 2.

Will be revised accordingly.

page20, lane393: Line break before: "First, local communities..."

Will be revised accordingly.

page22, line454: Line break before: "The fact that research...".

Will be revised accordingly.

page22, line470: Number should be (vi)

Will be revised accordingly.

page23, line492: Disclaimer: Please state who or what ICIMOD is.

ICIMOD is International Centre for Integrated Mountain Development. The full name will be used in the revised version of the manuscript.

**Citation**: https://doi.org/10.5194/nhess-2022-143-RC2

Thank you again for reviewing our manuscript!

On behalf of the collective of authors

Adam Emmer